# Effects of Meditation on Mental Health and Cardiovascular Balance in Caregivers

**DOI:** 10.3390/ijerph18020617

**Published:** 2021-01-13

**Authors:** Lourdes Díaz-Rodríguez, Keyla Vargas-Román, Juan Carlos Sanchez-Garcia, Raquel Rodríguez-Blanque, Guillermo Arturo Cañadas-De la Fuente, Emilia I. De La Fuente-Solana

**Affiliations:** 1Research Group CTS1068, Andalusia Research Plan, Junta de Andalucía, Hospital Universitario Virgen de las Nieves, 18014 Granada, Spain; keyvarom@ugr.es (K.V.-R.); jsangar@ugr.es (J.C.S.-G.); raquel.rodriguez.blanque.sspa@juntadeandalucia.es (R.R.-B.); 2Department of Nursing, School of Health Sciences, University of Granada, 18016 Granada, Spain; gacf@ugr.es; 3Spanish Education Ministry Program FPU16/01437, Methodology of Behavioral Sciences Department, School of Psychology, University of Granada, 18071 Granada, Spain; 4San Cecilio University Hospital, 18016 Granada, Spain; 5Methodology of Behavioral Sciences Department, School of Psychology, University of Granada, 18071 Granada, Spain; edfuente@ugr.es

**Keywords:** meditation, vagal nerve activity, high-burden caregivers, mental health

## Abstract

Background: Caring for a loved one can be rewarding but is also associated with substantial caregiver burden, developing mental outcomes and affecting happiness. The aim of this study was to determine the effects of a four-week, 16-h presential meditation program on physiological and psychological parameters and vagal nerve activity in high-burden caregivers, as compared to a control group. Methods: A non-randomized repeated-measures controlled clinical trial was conducted. Results: According to the ANCOVA results, the global happiness score (F = 297.42, *p* < 0.001) and the scores for all subscales were significantly higher in the experimental group than in the control group at 5 weeks. Anxiety levels were also significantly reduced in the experimental group (F = 24.92, *p* < 0.001), systolic (F = 16.23, *p* < 0.001) and diastolic blood (F = 34.39, *p* < 0.001) pressures, and the resting heart rate (F = 17.90, *p* < 0.05). HRV results revealed significant between-group differences in the HRV Index (F = 8.40, *p* < 0.05), SDNN (F = 13.59, *p* < 0.05), and RMSSD (F = 10.72, *p* < 0.05) in the time domain, and HF (F = 4.82 *p* < 0.05)) in the frequency domain, which were all improved in the experimental group after the meditation program. Conclusions: Meditation can be a useful therapy to enhance the mental health and autonomic nervous system balance of informal caregivers, improving symptoms of physical and mental overload.

## 1. Introduction

The term informal caregivers is applied to family members or close relatives providing partial or full care to dependent individuals with difficulties in self-care, facilitating their well-being and helping them to perform different tasks and activities [1,2]. Caring for a loved one can be rewarding but can also be associated with significant caregiver burden. In broad terms, this refers to stress due to caregiving that arises from an interplay among various predisposing factors, including contextual circumstances, direct primary stressors, indirect secondary stressors, and appraisal [3]. The physical health of caregivers can be a predictor of both care burden and depression, given that caregivers with poorer health might perceive a greater burden and be more prone to depression after a long period of caregiving; it is therefore necessary to adjust results for burden and depression, in order to establish their relative impact [4,5,6].

Depression is reported in at least one-third of caregivers of persons with dementia, a higher prevalence than that observed in the general population or in the caregivers of persons with other physical or psychological diseases [7]. Researchers also described a correlation between the anxiety of caregivers over an uncertain future and burnout [8]. However, although there is some evidence that caregiver burden can generate unhappiness [9], there is no research on the effect of interventions on their self-perception of happiness. Besides the psychological consequences of caregiver burden, it is associated with worse self-care, including a less healthy diet, fewer preventive medical visits, and lower physical activity levels, increasing the risk of cardiovascular and other diseases [10]. It was also proposed that stress generated by the caregiver–patient relationship increases the likelihood of coronary disease [11].

All disorders derived from the physical and emotional overload of caregivers can produce an imbalance of the autonomic nervous system (ANS), formed by the sympathetic and parasympathetic nervous systems [12]. The main component of the parasympathetic nervous system is the vagus nerve, which regulates mood status, immune response, digestion, and heart rate, among other key functions. Heart rate variability (HRV) results from interaction between the ANS and the cardiovascular system and reflects vagal activity, serving as a noninvasive biomarker of health and emotional regulation [13]. HRV is measured by calculating the time period between consecutive R waves (RR interval). Differences between successive heartbeats can be established with reference to time (time-domain analysis) or frequency (frequency-domain analysis) [14].

An increase in HRV is linked to adaptive emotional regulation strategies [15,16]. Practices that focus on the interactions among brain, body, mind, and behavior, such as yoga, are reported to improve the well-being and cardiac balance in informal caregivers [17,18]. “Meditation” is another ancient approach to the cultivation of well-being [19]. Numerous types of meditation are taught, often derived from different Eastern religious and spiritual traditions, but they all emphasize the regulation of attention and emotion, consciousness of the body, and self-awareness [20]. Mediation can be practiced by focusing on an object or event (Focused Attention), or in a more advanced modality, without recourse to this tactic (Open Monitoring) [21]. Meditation is reported to offer multiple psychoneurophysiological benefits, including reduction in stress and inflammation [22]; increase in attentional networks at the neural level [23], enhancement of explicit functions of parts of the right hemisphere [24], and the alleviation of psychological distress in cancer patients [25]. The American Heart Association considers meditation as an adjunct to guideline-directed cardiovascular risk reduction [26].

There is little published evidence on the efficacy of intervention programs to improve the well-being of informal caregivers. To our best knowledge, there are no previous studies on the effect of a short meditation program on the cardiovascular balance and mental health of high-burden informal caregivers. We hypothesized that a four-week program offering 16 h of meditation would improve the psychological outcomes and cardiovascular balance in informal caregivers compared to the control group, producing an increase in happiness and HRV and a reduction in anxiety and depression. The objective of the present study was to determine the effects of a four-week presential meditation program of 16 h on physiological and psychological parameters and vagal nerve activity in high-burden caregivers, comparing the results with those of controls not receiving this program.

## 2. Materials and Methods

### 2.1. Study Design

A non-randomized repeated-measures controlled clinical trial was conducted, allocating participants to an intervention or control group according to their ability or inability to attend all presential sessions in the program. It was not possible to randomly assign participants for ethical reasons, because it would mean denying the program to some caregivers requesting it. The study is registered at ClinicalTrials.gov (NCT04570826).

### 2.2. Setting and Selection of Participants

Relevant associations in the city and province of Granada city were contacted by the researchers in person or by telephone, to recruit informal caregivers for the study. Inclusion criteria for participation were care for at least two years of a dependent family member or close relative living in the same dwelling; no receipt of any economic remuneration, and a caregiver burden > 55 points on the Zarit Burden Scale [27]. Each item in this scale was scored on a 5-point Likert scale (0 = never to 4 = nearly always), yielding a total score ranging from 0 to 88. No consensus was established on the cutoff score for a high burden; however, after a review of the literature, a value of 55 points was selected. Exclusion criteria were history of cardiovascular disease; and previous experience in mind–body practices. Written informed consent was obtained from all participants in the study, which was approved by the local research ethics committee (CEI-GR C-9) and followed the principles of the Declaration of Helsinki. Convenience sampling was used to assign caregivers to the control or experimental groups. Figure 1 depicts the flow of participants through the study. A single researcher (L.D.-R.) contacted participants by telephone to collect data on their medical history and demographic characteristics, including age, gender, ethnicity, marital status, educational level, occupation, alcohol and smoking habits, menopause status, time performing caring activity, relationship with care-receiver, weight, and height.

### 2.3. Control Group

At the time of their allocation to the control group, these caregivers received a sheet that summarized the theory and practice of meditation and outlined its scientific basis.

### 2.4. Experimental Group

The experimental group participated in a one-month meditation program of eight 2 h sessions.

### 2.5. Meditation Training Program

A meditation trainer with more than 10 years of experience ran a Focused Attention Meditation program in a room at the School of Health Sciences of the University of Granada. Over a four-week period, two 2 h sessions were conducted each Friday afternoon, with a 30 min interval between them. In brief, the aims were to learn a comfortable posture when sitting or lying down; to focus attention on the breathing and allow distractions to come and go naturally without judging.

Each two-hour session started with a 15 min class on the scientific evidence supporting the exercises, followed by 25 min of exercises to promote mobility, flexibility, balance, strength, and endurance (Downward Dog, Child’s Pose, High Lunge, Triangle Pose, Mountain Pose, Cat/Cow Pose, Bridge Pose, Seated Forward Bend, Tree Pose, Pigeon Pose) and then by 25 min of costal, diaphragmatic, and clavicular breathing exercises to develop lung capacity (Stimulating Breath or Bellow Breath, Relaxing Breathing Exercise, Counting the Breath), with a final 25 min of body awareness exercises, observing sensations, thoughts and perceptions, with gratitude and compassion (Body Scan Meditation, Progressive Muscle Relaxation, Visualization, Conscious Observation). Although free to repeat the exercises at home, no participant reported being able to do so, citing time constraints.

#### Sample Size Calculation

The EPIDAT 4.1 software (Xunta de Galicia, Spain) was used to estimate the sample size for a statistical power of 80% with α = 0.05, based on previously published data [28]. A minimum sample size of 23 participants per group was calculated.

### 2.6. Outcome Measures

All outcome measures were determined before (week 0) and after (week 5) the meditation program. At both time points, data were gathered by a single researcher (L.-D.R.).

Happiness level, measured as scores on the validated self-administered Lima happiness questionnaire [29]. It contains 27 items grouped into 4 subscales (positive sense of life, satisfaction with life, personal fulfillment, and joy of life), with responses on a 5-point Likert scale (1 = strongly agree, 2 = agree, 3 = neither agree nor disagree, 4 = disagree, and 5 = strongly disagree). Total scores of 27–87 = very low, 88–95 = low, 96–110 = medium, 111–118 = high, and 119–135 = very high levels of happiness.

Hospital Anxiety and Depression Scale (HADS) score: This validated self-administered instrument is designed for general medical outpatients, to detect the possible presence of anxiety and depression. It contains 7 items for anxiety and 7 for depression, with responses on a 4-point Likert scale (0–3), in relation to feelings and emotions during the previous week [30].

Short-term HRV: The method published by Kappussami (2020) [31] was used, following recommendations of the Task Force of the European Society of Cardiology and North American Society of Pacing and Electrophysiology [14]. First, participants lay in a supine position in a quiet room (at 22–25 °C) for 10 min of rest with normal breathing paced by a metronome at 0.2 Hz. Next, ECG signals were acquired for 5 min using a Holter monitor with modified lead II channel system (Norav Holter NR302, Braemar, Brunsville, MN, USA). HRV was calculated from the ECG records as the time interval between consecutive heartbeats (RR interval). The following parameters were determined in the time domain—standard deviation of mean normal-to-normal (NN) interval (SDNN), square root of the mean squared differences of successive NN intervals (RMSSD), and number of all NN intervals divided by the maximum of all NN intervals (HRV index). The following spectral components were determined in the frequency domain—low-frequency (LF) band (0.04–0.15 Hz), as a measure of sympathetic and parasympathetic activities; high-frequency (HF) band (0.15–0.40 Hz), associated with vagal–parasympathetic activity; and LF/HF ratio, indicating the sympathovagal balance. The spectral analysis was performed with the NH301-4 software (Norav, version 2.70), using fast Fourier transform algorithms. The sampling rate was 256 samples per second and the frequency filter was set at 0.05 to 60 Hz. Due to the low sampling rate, an interpolation algorithm was used to improve R-peak detection and the frequency filter was set at 0.05–60 Hz.

Blood pressure/heart rate: Blood pressure and heart rate were measured at 0 and 5 weeks between 9 a.m. and 12 noon, using an Omron HEM-7320-Z validated automatic oscillometer, placing the cuff at 2 cm above the elbow and instructing the participant to not speak or move during the measurement.

### 2.7. Statistical Analysis

IBM-SPSS 26.0 was used for the statistical analysis. Results were expressed as means with standard deviation for continuous variables and percentages with 95% confidence intervals for categorical variables. Between-group differences at baseline were analyzed using *t*-tests for continuous variables and chi-square tests for categorical variables. After verifying the normality of the data distribution with the Kolmogorov–Smirnov test, an analysis of covariance (ANCOVA) was performed with time (pre, post intervention) as a within-subjects variable, intervention (meditation, control group) as between-subjects variable, and baseline levels of variables as covariates. The Bonferroni correction was used for post-hoc pairwise comparisons and <0.05 was considered statistically significant. The hypothesis of interest was intervention × time interaction. When an interaction was found, the inter-group effect size was calculated according to eta squared (η^2^), i.e., the proportion of variance in the dependent variable that is attributable to the factor in question. A η^2^ of 0.01 signifies a small effect, η^2^ of 0.06 a medium effect, and η^2^ of 0.14 a large effect [32].

After verifying the normality of the data distribution with the Kolmogorov–Smirnov test, an analysis of covariance (ANCOVA) was performed, with pre-post differences as the dependent variables, group as the fixed factor (Meditation and Control), and baseline levels of variables as covariates. The Bonferroni correction was used for post-hoc pairwise comparisons, and <0.05 was considered to be statistically significant.

## 3. Results

Forty informal caregivers who met the eligibility criteria were initially enrolled in the study. The program was not completed by three participants, leaving a final study sample of 37 caregivers, 28 females, and 9 males, with a mean (SD) age of 44.03 (7.30) years, mean height of 165.72 (6.85) cm, mean weight of 69.64 (13.43) kg and, therefore, a mean BMI of 25.26 (4.09) kg/m^2^. All participants except one were Caucasian, 73.7% were married, >50% had completed a higher education, 78.4% were employed, 56.8% were nonsmokers, 60% did not consume alcohol, 70.2% were caring for a relative for >6 years, and 54.1% were parents of the care-receivers. The only statistically significant difference in the above variables between the experimental (n = 19) and control (n = 18) groups was in educational level, with higher education being completed by 31.6% of the experimental group versus 72.2% of controls (Table 1).

There was no significant between-group difference in any variable at baseline. According to the ANCOVA results, the global happiness score (F = 297.42, *p* < 0.001, η^2^ = 0.89) and the scores for all following subscales were significantly higher in the experimental group than in the control group at 5 weeks—positive sense of life (F = 74.61, *p* < 0.001, η^2^ = 0.68), satisfaction with life (F = 111.62, *p* < 0.001, η^2^ = 0.76), personal realization (F = 41.64, *p* < 0.001, η^2^ = 0.54), and happiness of living (F = 234.57, *p* < 0.001, η^2^ = 0.87). Intergroup effect sizes were large for all subscales.

Anxiety levels were also significantly reduced in the experimental group, with a large effect size (F = 24.92, *p* < 0.001, η^2^ = 0.41) (Figure 2). An improvement in depression levels in the experimental versus control group did not reach statistical significance (F = 1.75, *p* > 0.05).

HRV results revealed significant between-group differences with a large effect size in the HRV Index (F = 8.40, *p* < 0.05, η^2^ = 0.19), SDNN (F = 15.59, *p* < 0.05, η^2^ = 0.28), and RMSSD (F = 10.72, *p* < 0.05, η^2^ = 0.23), in the time domain, and a medium effect size in the HF (F = 4.82 *p* < 0.05, η^2^ = 0.09) in the frequency domain, which were all improved in the experimental group, after the meditation program (Table 2). Significantly decreased systolic (F = 50.68, *p* < 0.001, η^2^ = 0.31) and diastolic blood (F = 38.14, *p* < 0.001, η^2^ = 0.49) pressures and resting heart rate (F = 12.62, *p* < 0.05, η^2^ = 0.33) with a large effect size were also observed in the experimental group at 5 weeks. No covariates had an influence in all these results.

## 4. Discussion

To the best of our knowledge, this is the first controlled clinical trial to demonstrate an improvement in the mental health and cardiovascular balance of high-burden informal caregivers after a one-month meditation program of eight 2 h sessions, in comparison to a control group. After the program, the caregivers evidenced a decrease in blood pressure, resting heart rate, and anxiety, and an increase in happiness score and HRV. No statistically significant between-group differences were observed in the LF band or the LF/HF ratio, or in depression levels.

These data contribute evidence on the effectiveness of complementary therapies to enhance the mental health of healthy individuals by promoting well-being and improving their psychological function [31]. With regard to the impact on anxiety, a recent meta-analysis supported the usefulness of yoga or meditation as a complementary or solo treatment of anxiety or depression [33]. It was previously found that a 12-month yoga program or an 8-week meditation programs could reduce anxiety and depression and diminish physical and psychological distress [17,18,34].

Besides reducing anxiety levels, the participants receiving the meditation program experienced an improvement in happiness, as measured by the Lima Scale, from a very low initial score to a medium score. They reported a more positive perception of their sense of life, satisfaction with life, personal realization, and happiness of living. Klamut (2002) [35] described happiness as a state of inner peace and satisfaction with life, characterized by benevolence towards oneself and others; sensitivity to the beauty of nature, culture, and art, and a harmonious coexistence with the environment. In a study of 46 caregivers of patients with Alzheimer’s disease, Danukalov et al. (2017) [28] observed that an eight-week yoga and meditation program improved their quality of life and vitality and enhanced their capacity for attention and self-compassion, which demonstrated potential benefits for mental and physical health [36]. Other authors closely associated genuine and enduring happiness with compassion and empathy, related to a selfless mode, based on the dissolution of the limits perceived between the body and the rest of the world [37,38].

In the present study, higher scores for Satisfaction with Life were correlated with a higher HRV, and the presence of anxiety correlated with a lower HRV, demonstrating a positive relationship between mental well-being and HRV. In this regard, the HRV was influenced by functions of the prefrontal cortex, whose activity are associated with long-term happiness [39]. In the time domain, SDNN, RMSSD, and HRV index values were significantly higher after the meditation program, in comparison to controls. Similar findings were reported after a six-month yoga breathing program in healthy adolescents [31] and after myofascial therapy or Reiki sessions in breast cancer survivors with cancer-related fatigue (increased SDNN and RMSSD), in stress responders (increased SDNN), and in burned-out healthcare professionals (increased SDNN) [15,16]. In the frequency domain, HF band values were significantly higher in caregivers who underwent the meditation program, indicating activation of the parasympathetic nervous system, as previously observed, after manual treatment or music therapy [40,41]. A meditation session in healthy individuals [42] and an eight-week yoga program in women with high depressive symptoms, Chu (2015) [43] were able to exert positive effects on sympathovagal balance, increasing the HF and reducing the LF and LF/HF ratio. However, the comparison of HRV findings among studies was hampered by differences in measurement techniques and study parameters.

The positive effects of meditation on HRV, blood pressures, and HR were previously described [15,44,45]. However, this was the first controlled clinical case to demonstrate that the mental health and cardiovascular balance of high-burden caregivers could be improved by a meditation program, with a significant reduction in the resting heart rate and in the systolic and diastolic blood pressures.

Study limitations included the small sample size and the lack of randomization, which reduced the statistical power and the possibility of extrapolating results. In addition, the educational level was higher in the control group, and this difference should be avoided in future studies. It would also be of interest to study whether the effectiveness of the intervention differed according to the burden of caregivers or the presence or not of dementia in their charges. The program was also relatively short (4 weeks), and a larger sample size and longer program might confirm an improvement in depression and certain HRV parameters that did not reach statistical significance in the present investigation.

## 5. Conclusions

A short meditation program might be a useful tool for primary healthcare professionals to enhance the mental health and cardiovascular balance of informal caregivers, improving their happiness and psychological symptoms. Further clinical trials are warranted to verify the efficacy of mind-body interventions to improve the quality of life of informal caregivers.

## Figures and Tables

**Figure 1 ijerph-18-00617-f001:**
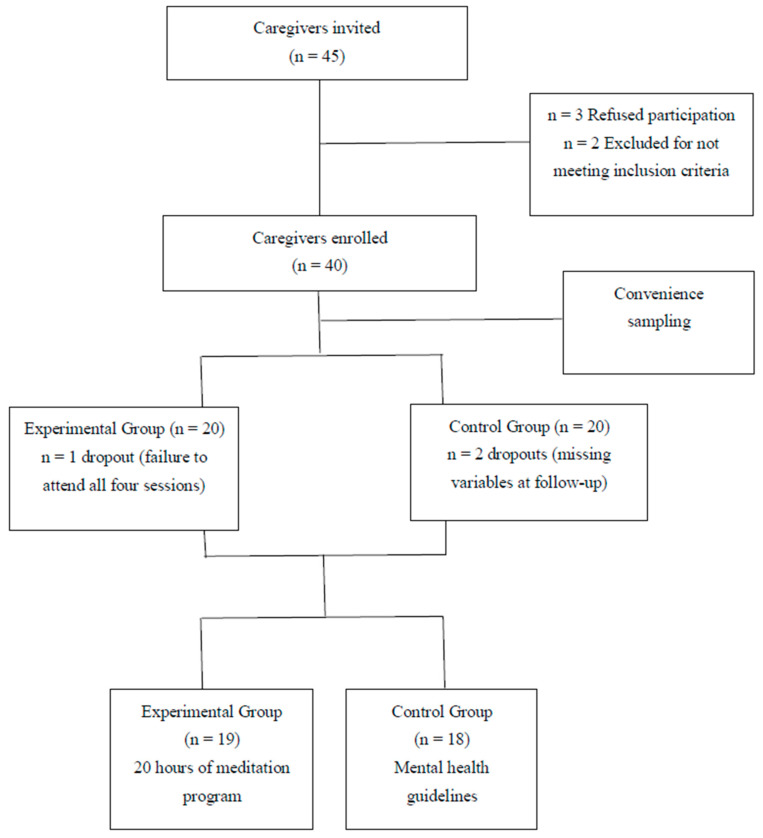
Flow of participants.

**Figure 2 ijerph-18-00617-f002:**
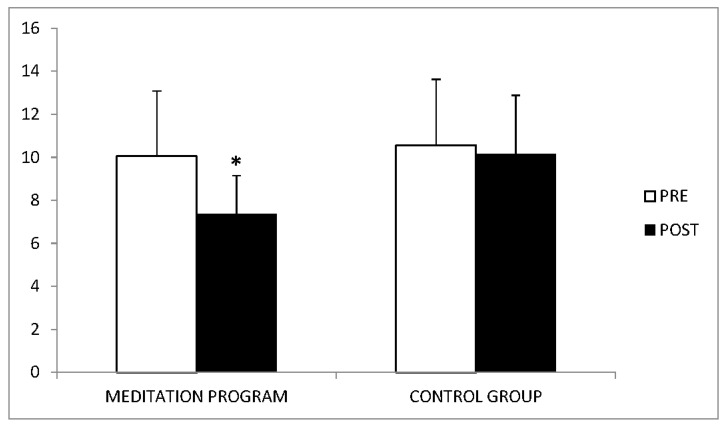
Comparison of anxiety values between before and after treatments. * *p* < 0.05.

**Table 1 ijerph-18-00617-t001:** Caregiver characteristics and comparisons between study groups.

Variables	Meditation Program (*n* = 19)	Control Group (*n* = 18)	*p*
**Age (y) Mean (SD)** **	44.42 ± 8.17	43.61 ± 6.46	
	(33–64)	(34–56)	0.74
**Gender (%)** *			
Female	73.7	77.8	
Male	26.3	22.2	0.77
**Ethnicity (%)** *			
Caucasian	94.7	100	
Black	5.3	0.0	
Gypsy	0.0	0.0	
Arab	0.0	0.0	0.32
**Marital status (%)** *			
Single	5.3	0.0	
Married	73.7	72.2	
Divorced	15.8	16.7	
Widow	5.3	11.1	0.71
**Educational level (%)** *			
Primary studies	26.3	0.0	
Secondary studies	42.1	27.8	
Higher Education	31.6	72.2	0.01 *
**Occupational status (%)** *			
Homemaker	31.6	11.1	
Employed	68.4	88.9	
Unemployed	0.0	0.0	
Retired	0.0	0.0	0.13
**Smoking status (%)** *			
Non-smoker	57.9	55.6	
Smoker	15.8	22.2	
Ex-smoker	26.3	22.2	0.87
**Alcohol status(%)** *			
Don’t consume	68.4	50.0	
Consume monthly	10.5	22.2	
Consume weekly	21.1	27.8	
Consume daily	0.0	0.0	0.47
**Menopausal status (%)** *			
NO	84.2	94.4	
YES	15.8	5.6	0.31
**Duration of caring (%)** *			
1–5 years	26.3	33.0	
6–10 years	47.4	38.9	
>11 years	26.3	27.8	0.85
**Caregiver relationship with care receiver (%)** *			
Partner	5.3	5.6	
Parent	52.6	55.6	
Child	42.1	38.9	0.98
**Weight (Kg) Mean (SD)** **	70.60 ± 12.78	68.63 ± 14.38	
	(53–110)	(56–105)	0.66
**Height (cm) Mean (SD)** **	165.78 ± 7.08	165.66 ± 6.80	
	(155–180)	(158–179)	0.95
**Body Mass Index Mean** ** **(Kg/m^2^) (SD)**	25.73 ± 4.77	24.76 ± 3.30	
	(20.20–39.56)	(21.01–32.77)	0.48

**Note:** Values are expressed as means  ±  standard deviation (95% confidence interval). Chi-square test * and Student *t*-test ** for between-group comparisons; * *p* < 0.05.

**Table 2 ijerph-18-00617-t002:** Comparison of outcomes between before and after treatments.

Outcomes	Meditation Program (*n* = 19)	Control Group (*n* = 18)	F
**Heart rate (beat * min^−1^)**			
Baseline	75.05 ± 10.01	73.89 ± 6.41	
Post-treatment	63.42 ± 1035	72.72 ± 6.93	17.90 *
**Systolic Pressure Blood (mm Hg)**			
Baseline	127.21 ± 16.00	128.16 ± 9.83	
Post-treatment	115.63 ± 14.01	127.22 ± 10.54	16.23 **
**Diastolic Pressure Blood (mm Hg)**			
Baseline	74.05 ± 9.57	73.16 ± 3.22	
Post-treatment	67.57 ± 7.50	72.05 ± 3.36	34.39 **
**Heart Rate Variability**			
**SDNN**			
Baseline	45.08 ± 28.83	52.47 ± 16.04	
Post-treatment	90.33 ± 50.65	55.65 ± 16.42	13.59 *
**RMSSD**			
Baseline	41.78 ± 30.16	49.64 ± 17.39	
Post-treatment	91.04 ± 63.81	54.58 ± 17.85	10.72 *
**HRV índex**			
Baseline	4.93 ± 2.56	6.02 ± 1.55	
Post-treatment	6.72 ± 2.70	5.91 ± 1.34	8.40 *
**LF**			
Baseline	171.08 ± 60.09	137.82 ± 41.97	
Post-treatment	169.00 ± 69.33	144.64 ± 46.41	0.25
**HF**			
Baseline	141.86 ± 61.25	140.88 ± 31.49	
Post-treatment	167.34 ± 65.13	137.90 ± 31.48	4.82 *
**LF/HF Ratio**			
Baseline	1.39 ± 0.76	1.04 ± 0.45	
Post-treatment	1.38 ± 1.44	1.10 ± 0.52	0.06

**Note:** ANCOVA for comparisons between interventions * *p* < 0.05 ** *p* < 0.001. SDNN = standard deviation of the normal-to-normal interval; RMSSD = root mean square of successive differences; LF = low frequency; HF = high frequency; and ANCOVA = analysis of covariance.

## Data Availability

Data available in a publicly accessible repository. The data presented in this study are openly available in [repository FigShare] at [doi], reference number [https://figshare.com/s/4a711026df2ce16e21dd reference number].

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
