# Peer review of "Effects of Meditation on Mental Health and Cardiovascular Balance in Caregivers"

_ijerph, 2021, doi:10.3390/ijerph18020617_

Round 1

Reviewer 1 Report

The first impression after reading this paper is more than positive. The goal of this review is clearly described.
As rightly pointed out by the authors, I agree that “..., this is the first controlled clinical trial to demonstrate an improvement in the mental health and cardiovascular balance of high-burden informal caregivers after a one month meditation program”

Just could you explain why did you choose this kind of "dosage" of meditation - two 2-hour sessions were conducted each Friday afternoon, with a 30-min interval between them.

Author Response

Response letter Manuscript ijerph-1039938

EFFECTS OF MEDITATION ON MENTAL HEALTH AND CARDIOVASCULAR BALANCE IN CAREGIVERS

Lourdes Díaz-Rodríguez, Keyla Vargas-Román, Juan Carlos Sanchez-Garcia, Raquel Rodríguez-Blanque,  Guillermo Arturo Cañadas-De la Fuente and Emilia I. De La Fuente-Solana.

We would like to thank the reviewers for their insightful comments, which have greatly enhanced the overall quality and readability of the manuscript. Below please find a list of revisions and responses to each of the reviewer’s comments. We have highlighted all changes in the manuscript in yellow.

Revision requirements

The first impression after reading this paper is more than positive. The goal of this review is clearly described. As rightly pointed out by the authors, I agree that “..., this is the first controlled clinical trial to demonstrate an improvement in the mental health and cardiovascular balance of high-burden informal caregivers after a one month meditation program”.

 Comment 1: Just could you explain why did you choose this kind of "dosage" of meditation - two 2-hour sessions were conducted each Friday afternoon, with a 30-min interval between them.

Response: According to our review of the literature, meditation programs most frequently last for eight weeks and offer one-hour sessions two or three times per week, i.e., a total of 16-30 hours of meditation sessions (Chu 2015, Danukalov 2017). However, given the very limited time available to caregivers for this activity, we decided that it should last for four weeks and only require attendance on one day per week but should be more intense, offering two two-hour sessions separated by a 30-min rest period, i.e., a total of 16 hours of meditation sessions.

Reviewer 2 Report

  1. The introduction generally reads well, is informative and provides a context for the study. One point would help the clarity of this section if the authors could provide one paragraph reviewing the limitations of that current evidence that this study will help to address, and stating explicit hypotheses at the close of the introduction.
  2. Line 79: Suggest adding a sub-heading "Study design". Also the study design needs more details and clarity.
  3. Line 97-98: Please include more details about the "Control Group".
  4. Results are not well organized. Each table should be separate and include a brief description of the results being presented.
  5. Figure 1 should be moved to method section.
  6. Line 183: Suggest deleting this sub-heading.
  7. Line 189-190: Chi-square and t-test should be mentioned and described in statistical analysis section.
  8. Table 3: Abbreviations should be defined in the footnote.
  9. Conclusions do not appear to be supported by the results. The authors should also mention the implications of the study in this section.
  10. Authors should follow the journal guidelines for references. Also, reference #11 is old-please update.

Author Response

EFFECTS OF MEDITATION ON MENTAL HEALTH AND CARDIOVASCULAR BALANCE IN CAREGIVERS

Lourdes Díaz-Rodríguez, Keyla Vargas-Román, Juan Carlos Sanchez-Garcia, Raquel Rodríguez-Blanque,  Guillermo Arturo Cañadas-De la Fuente and Emilia I. De La Fuente-Solana.

We would like to thank the reviewers for their insightful comments, which have greatly enhanced the overall quality and readability of the manuscript. Below please find a list of revisions and responses to each of the reviewer’s comments. We have highlighted all changes in the manuscript in yellow.

Comments and Suggestions for Authors 2

Comment 1: The introduction generally reads well, is informative and provides a context for the study. One point would help the clarity of this section if the authors could provide one paragraph reviewing the limitations of that current evidence that this study will help to address, and stating explicit hypotheses at the close of the introduction.

Response: We have added a paragraph on the limitations of current evidence, followed by a statement of the explicit hypotheses of our study at the end of the Introduction, as requested:

“There is little published evidence on the efficacy of intervention programs to improve the well-being of informal caregivers. To our best knowledge, there has been no previous study of the effect of a short meditation program on the cardiovascular balance and mental health of high-burden informal caregivers. We hypothesized that a four-week program offering 16 hours of meditation would improve psychological outcomes and cardiovascular balance in informal caregivers compared with a control group, producing an increase in happiness and HRV and a reduction in anxiety and depression.

”.

Comment 2: Line 79: Suggest adding a sub-heading "Study design". Also the study design needs more details and clarity.

Response: We have added the sub-heading and expanded the “Study design” paragraph as follows:

A non-randomized repeated-measures controlled clinical trial was conducted, allocating participants to an intervention or control group according to their ability or inability to attend all presential sessions in the program. It was not possible to randomly assign the participants for ethical reasons, because it would mean denying the program to some caregivers requesting it. The study was registered at ClinicalTrials.gov (NCT04570826).

Comment 3: Line 97-98: Please include more details about the "Control Group".

Response: This has been done, as follows:

At the time of their allocation to the control group, these caregivers received a sheet that summarized the theory and practice of meditation and outlined its scientific basis.

Comment 4: Results are not well organized. Each table should be separate and include a brief description of the results being presented.

Response: This has been done.

Comment 5: Figure 1 should be moved to method section.

Response: This change has been made.

Comment 6: Line 183: Suggest deleting this sub-heading.

Response: Done

Comment 7: Line 189-190: Chi-square and t-test should be mentioned and described in statistical analysis section.

Response: We have added the following sentence in the statistical analysis section:

Between-group differences at baseline were analyzed using t-tests for continuous variables and chi-square tests for categorical variables

Comment 8: Table 3: Abbreviations should be defined in the footnote.

Response: Done

Comment 9: Conclusions do not appear to be supported by the results. The authors should also mention the implications of the study in this section.

Response: This section has been rewritten accordingly:

A short meditation program may be a useful tool for primary health care professionals to enhance the mental health and cardiovascular balance of informal caregivers, improving their happiness and psychological symptoms.

 Comment 10: Authors should follow the journal guidelines for references. Also, reference #11 is old-please update.

Response: References now comply with the journal guidelines. We would prefer to preserve reference 11 as a key document.

Reviewer 3 Report

The manuscript presents a novel investigation that relate mental health and cardiovascular balance of high-burden informal caregivers using a meditation-based intervention program.

In paragraph 1 line 4 it is important to justify the relationship between well-being and activities; Meléndez, J. C., Tomás, J. M., & Navarro, E. (2011). Everyday life activities and well-being: Their relationships with age and gender in the elderly. Annals of Psychology, 27(1), 164-169 confirm the relationship between activities of daily living with dimensions of well-being. Please include this reference.

Also, in relation to the health of caregivers in older adults (line 38 and following) two papers published in the journal can offer interesting and updated information to complete their background. Please include these references. Marfil-Gómez, R., Morales-Puerto, M., León-Campos, Á., Morales-Asencio, J. M., Morilla-Herrera, J. C., Timonet-Andreu, E., ... & García-Mayor, S. (2020). Quality of Life, Physical and Mental Health of Family Caregivers of Dependent People with Complex Chronic Disease: Protocol of a Cohort Study. International Journal of Environmental Research and Public Health, 17(20), 7489.

Ruisoto, P., Ramírez, M., Paladines-Costa, B., Vaca, S., & Clemente-Suárez, V. J. (2020). Predicting Caregiver Burden in Informal Caregivers for the Elderly in Ecuador. International Journal of Environmental Research and Public Health, 17(19), 7338.

In relation to the consequences of caring for health, it could be interesting for the authors to differentiate between the different types of subjects / patients or dependent people who are cared for. If the authors prefer to focus on older adults (line 42), you could group the consequences according to the presence or not of a dementia diagnosis in the subject to whom care is provided. Although there are common aspects, there are also differences in the consequences for the main caregiver. Please, clarify this aspect.

In the method section, when the authors indicate the inclusion criteria, you report “care for at least two years of a dependent family member or close relative living in the same dwelling; and a caregiver burden of >55 points on the Zarit Burden Scale”. It is possible that although you have not indicated some criteria, you have actually applied them, for example: (a) identify themselves as the primary relative providing informal care to an older adult; (b) dedicate at least XX day a week to caring for the family member; (c) not receive any economic remuneration; (d)… If this is the case, please include them in such a way that the inclusion criteria are better defined and the population object of your intervention and study can be better differentiated.

More information about the Meditation Training Program is needed. It would make it easier to have a broader view of the program, objectives, activities, etc. Please expand the information so that your intervention can be replicated.

Although you do not include the Zarit Burden Scale in the outcome subsection, it would be important that you provide general information on the scale; You include this instrument as a main element in the inclusion criteria, indicating that those subjects with a score> 55 participate in the study; It is important to know the type of measurement used, its maximum range, if there is a cut-off point, etc. From my point of view, I think it would be very interesting to know if the burden of the subjects to whom the intervention is applied shows a reduction in their scores. This should be a core point of your work, especially when it is an inclusion criterion.

Please include a reference to the Lima happiness questionnaire; validation, adaptation, version used, reliability, etc. (Have you used the original version of Alarcon 2006?)

The authors could try applying repeated measures of the SPSS; This is an idea; it is not a modification that they have to make since the use of baseline levels of variables as covariates is correct; you can do it or not if you think it improves your results. The repeated measures can be applied with Bonferroni correction and it offers them for each of the measures the different meanings, in the pre, in the post and intragroup; in addition, it minimizes type I error.

Regarding the results, it would be advisable to include the effect size of their results.

The p value of 0.000 does not exist will be <0.001. In my opinion, the authors should offer the results in text format, not including two tables, although the final decision is yours.

I observe a high difference of means in the base line for the measures of: HEART RATE VARIABILITY SDNN (41.78 vs 49.64), RMSSD (4.93 vs 6.02), LF (171.08 vs 137.82); if there are differences in the baseline, these variables should not be compared since the groups have different starting points and are not equal. Please review it and determine if you keep the result.

If as the authors point out "It was not possible to randomly assign the participants for ethical reasons", please remove the lack of randomization from the limitations

You could include as a limitation the differences observed between the groups in educational level; the control group has a significantly higher level; this variable should be controlled in future research.

Author Response

Response letter Manuscript ijerph-1039938

 EFFECTS OF MEDITATION ON MENTAL HEALTH AND CARDIOVASCULAR BALANCE IN CAREGIVERS

Lourdes Díaz-Rodríguez, Keyla Vargas-Román, Juan Carlos Sanchez-Garcia, Raquel Rodríguez-Blanque,  Guillermo Arturo Cañadas-De la Fuente and Emilia I. De La Fuente-Solana.

We would like to thank the reviewers for their insightful comments, which have greatly enhanced the overall quality and readability of the manuscript. Below please find a list of revisions and responses to each of the reviewer’s comments. We have highlighted all changes in the manuscript in yellow.

Comments and Suggestions for Authors 3

The manuscript presents a novel investigation that relate mental health and cardiovascular balance of high-burden informal caregivers using a meditation-based intervention program.

Comment 1: In paragraph 1 line 4 it is important to justify the relationship between well-being and activities; Meléndez, J. C., Tomás, J. M., & Navarro, E. (2011). Everyday life activities and well-being: Their relationships with age and gender in the elderly. Annals of Psychology, 27(1), 164-169 confirm the relationship between activities of daily living with dimensions of well-being. Please include this reference.

Response: We are grateful for this suggestion. This reference is now included in the revised manuscript.

Comment 2: Also, in relation to the health of caregivers in older adults (line 38 and following) two papers published in the journal can offer interesting and updated information to complete their background. Please include these references. Marfil-Gómez, R., Morales-Puerto, M., León-Campos, Á., Morales-Asencio, J. M., Morilla-Herrera, J. C., Timonet-Andreu, E., ... & García-Mayor, S. (2020). Quality of Life, Physical and Mental Health of Family Caregivers of Dependent People with Complex Chronic Disease: Protocol of a Cohort Study. International Journal of Environmental Research and Public Health, 17(20), 7489.

Ruisoto, P., Ramírez, M., Paladines-Costa, B., Vaca, S., & Clemente-Suárez, V. J. (2020). Predicting Caregiver Burden in Informal Caregivers for the Elderly in Ecuador. International Journal of Environmental Research and Public Health, 17(19), 7338.

Response: These have also been included in the revised manuscript.

Comment 3: In relation to the consequences of caring for health, it could be interesting for the authors to differentiate between the different types of subjects / patients or dependent people who are cared for. If the authors prefer to focus on older adults (line 42), you could group the consequences according to the presence or not of a dementia diagnosis in the subject to whom care is provided. Although there are common aspects, there are also differences in the consequences for the main caregiver. Please, clarify this aspect.

Response: This issue is indeed of interest but was outside the scope of our investigation. In fact, we had not elected to focus on “older adults”, and this misleading indication has been removed.  We have added the following to the Discussion:

“It would also be of interest to study whether the effectiveness of the intervention differs according to the burden of caregivers or the presence or not of dementia in their charges.”

Comment 4: In the method section, when the authors indicate the inclusion criteria, you report “care for at least two years of a dependent family member or close relative living in the same dwelling; and a caregiver burden of >55 points on the Zarit Burden Scale”. It is possible that although you have not indicated some criteria, you have actually applied them, for example: (a) identify themselves as the primary relative providing informal care to an older adult; (b) dedicate at least XX day a week to caring for the family member; (c) not receive any economic remuneration; (d)… If this is the case, please include them in such a way that the inclusion criteria are better defined and the population object of your intervention and study can be better differentiated.

Response: The inclusion criteria are now better defined, as requested:

Inclusion criteria for participation were: care for at least two years of a dependent family member or close relative living in the same dwelling; no receipt of any economic remuneration, and caregiver burden of >55 points on the Zarit Burden Scale [27].”

Comment 5: More information about the Meditation Training Program is needed. It would make it easier to have a broader view of the program, objectives, activities, etc. Please expand the information so that your intervention can be replicated.

Response: We have expanded the information on the program, as follows:

“Each two-hour session started with a 15-min class on the scientific evidence supporting the exercises, followed by 25 minutes of exercises to promote mobility, flexibility, balance, strength, and endurance (Downward Dog, Child's Pose, High Lunge, Triangle Pose, Mountain Pose, Cat/Cow Pose, Bridge Pose, Seated Forward Bend, Tree Pose, Pigeon Pose) and then by 25 minutes of costal, diaphragmatic, and clavicular breathing exercises to develop lung capacity (Stimulating Breath or Bellow Breath, Relaxing Breathing Exercise, Counting the Breath), with a final 25 minutes of body awareness exercises, observing sensations, thoughts and perceptions with gratitude and compassion (Body Scan Meditation, Progressive Muscle Relaxation, Visualization, Conscious Observation). Although free to repeat the exercises at home, no participant reported being able to do so, citing time constraints”.

Comment 6: Although you do not include the Zarit Burden Scale in the outcome subsection, it would be important that you provide general information on the scale; You include this instrument as a main element in the inclusion criteria, indicating that those subjects with a score> 55 participate in the study; It is important to know the type of measurement used, its maximum range, if there is a cut-off point, etc. From my point of view, I think it would be very interesting to know if the burden of the subjects to whom the intervention is applied shows a reduction in their scores. This should be a core point of your work, especially when it is an inclusion criterion.

Response: We have added the following information about the Zarit Burden Scale:

“…and a caregiver burden of >55 points on the Zarit Burden Scale [27]. Each item in this scale is scored on a 5-point Likert scale (0=never to 4= nearly always), yielding a total score ranging from 0 to 88. No consensus has been established on the cutoff score for a high burden; however, after a review of the literature, a value of 55 points was selected.”

Please also see our response to Comment 3 above.

Comment 6: Please include a reference to the Lima happiness questionnaire; validation, adaptation, version used, reliability, etc. (Have you used the original version of Alarcon 2006?)

Response: We used the original version, now cited as reference 29 (Alarcón, R. Desarrollo de una Escala Factorial para medir Felicidad. Interamerican Journal of Psychology, 2006; 40(1), 99-106), which has been validated in different Spanish-speaking populations (Rodriguez-Hernandez, G; Dominguez-Zacarias, G.; Escoto Ponce de Leon, M.C. Evaluación psicométrica de la escala de felicidad de Lima en una muestra mexicana. Univ. Psychol 2017; 16(4): 272-281 and Arraga Barrios, M.; Sánchez Villarroel, M. Validity and reliability of the Scale of Happiness from Lima in Venezuelan elderly people. Universitas psychologica, 2012; 11(2): 381-393).

Comment 7:

The authors could try applying repeated measures of the SPSS; This is an idea; it is not a modification that they have to make since the use of baseline levels of variables as covariates is correct; you can do it or not if you think it improves your results. The repeated measures can be applied with Bonferroni correction and it offers them for each of the measures the different meanings, in the pre, in the post and intragroup; in addition, it minimizes type I error.

Response: We applied repeated measures of the SPSS but observed no change in the results (data not shown).

Comment 8: Regarding the results, it would be advisable to include the effect size of their results.

Response: We have added the effect size (eta squared: η2).

The Statistical analysis section now includes the following:

The hypothesis of interest was intervention × time interaction. When an interaction was found, the inter-group effect size was calculated according to eta squared (η2), i.e., the proportion of variance in the dependent variable that is attributable to the factor in question. An η2 of 0.01 signifies a small effect, η2 of 0.06 a medium effect and η2 of 0.14 as large effect. [32].

Comment 9:

The p value of 0.000 does not exist will be <0.001. In my opinion, the authors should offer the results in text format, not including two tables, although the final decision is yours.

Response: The Happiness Table has been deleted. In the new Table 2, p-values are indicated by asterisks (* = P<0.05; ** = P<0.001)

Comment 10:

I observe a high difference of means in the base line for the measures of: HEART RATE VARIABILITY SDNN (41.78 vs 49.64), RMSSD (4.93 vs 6.02), LF (171.08 vs 137.82); if there are differences in the baseline, these variables should not be compared since the groups have different starting points and are not equal. Please review it and determine if you keep the result.

Response: There was no significant between-group difference in any variable at baseline. (SDNN: 45.08 vs 52.47, p=0.346; RMSSD: 41.78 vs 49.64, p=0.342; HRV index: 4.93 vs 6.02, p=0.127; LF: 171.08 vs 137.82, p=0.060; HF: 141.86 vs 140.88, p=0.952; and LF/HF ratio: 1.39 vs 1.04, p= 0.102)

 We have added this sentence in Results

“There was no significant between-group difference in any variable at baseline”.

Comment 10: If as the authors point out "It was not possible to randomly assign the participants for ethical reasons", please remove the lack of randomization from the limitations

Response: Done

Comment 11: You could include as a limitation the differences observed between the groups in educational level; the control group has a significantly higher level; this variable should be controlled in future research.

Response: The paragraph on study limitations now reads as follows:

“Study limitations include the small sample size, which reduced the statistical power and the possibility of extrapolating results. In addition, the educational level was higher in the control group, and this difference should be avoided in future studies. It would also be of interest to study whether the effectiveness of the intervention differs according to the burden of caregivers or the presence or not of dementia in their charges.”

Round 2

Reviewer 2 Report

Dear Authors,

I believe that the paper has been greatly improved by these revisions. One comment- the conclusion section should be expanded. The authors should make suggestions for further research in this section.

Author Response

We have added the following sentence to the conclusion section, as recommended:

 "Further clinical trials are warranted to verify the efficacy of
 mind-body interventions to improve the quality of life of informal
 caregivers."

Reviewer 3 Report

Thanks for including the recommendations

Author Response

We are grateful for the positive response.